# Infrared and Visible Image Fusion via Feature-Oriented Dual-Module Complementary

Yingmei Zhang [1] and Hyo Jong Lee [1,*]

Division of Computer Science and Engineering, CAIIT, Jeonbuk National University,
Jeonju 54896, Republic of Korea
* Correspondence: hlee@jbnu.ac.kr

**Abstract:** With the industrial demand caused by multi-sensor image fusion, infrared and visible image fusion (IVIF) technology is flourishing. In recent years, scale decomposition methods have led the trend for feature extraction. Such methods, however, have low time efficiency. To address this issue, this paper proposes a simple yet effective IVIF approach via a feature-oriented dual-module complementary. Specifically, we analyze five classical operators comprehensively and construct the spatial gradient capture module (SGCM) and infrared brightness supplement module (IBSM). In the SGCM, three kinds of feature maps are obtained, respectively, by introducing principal component analysis, saliency, and proposing contrast estimation operators considered the relative differences of contrast information covered by the input images. These maps are later reconstructed through pyramidal transformation to obtain the predicted image. The IBSM is then proposed to refine the missing infrared thermal information in the predicted image. Among them, we improve the measurement operators applied to the exposure modalities, namely, the gradient of the grayscale images (2D gradient) and well-exposedness. The former is responsible for extracting fine details, and the latter is meant for locating brightness regions. Experiments performed on public datasets demonstrate that the proposed method outperforms nine state-of-the-art methods in terms of subjective visual and objective indicators.

**Keywords:** infrared and visible image fusion; feature-oriented dual-module complementary; spatial gradient capture module; infrared brightness supplement module





## 1. Introduction

The purpose of multi-sensor image fusion is to extract and retain features from source images to generate a comprehensive image with abundant information. As one of the components of multi-sensor image pairs, infrared images reflect the thermal radiation information sensed from the scene but lack details regarding the scene; visible images contain considerably detailed information, but the positioning of infrared targets is severely controlled under harsh environments [1–3]. As one of the classic fusion approaches, infrared and visible image fusion (IVIF) has historically played a vital role. Fused images processed by this technique greatly benefit the subsequent advanced computer vision task, such as object detection [4–6], semantic segmentation [7,8], and pedestrian re-identification [9,10].

Over the past few years, several IVIF algorithms have been proposed and divided into two categories according to popularity: scale–transformation-based and deep-learning-based methods [1]. The methods in the first category generally entail four stages: pyramid transform, wavelet transform, edge-preserving filter, and hybrid multiscale filter decomposition. Bulanon et al. [11] proposed an IVIF method based on Laplacian pyramid transformation and fuzzy logic to detect fruits and obtained improved detection results. Zhan et al. [12] proposed a discrete-wavelet-transform-based IVIF method based on two fusion rules to obtain better performance. Meng et al. [13] proposed an IVIF method based on non-subsampled contourlet transform (NSCT) and object region detection; this method

first locates the infrared target region and then combines the sub-images decomposed by the NSCT to obtain the final fused image with good retention of targets and details. Hu et al. [14] proposed an IVIF method that uses a guided filter to decompose input images to obtain two sub-layers; this method also combines the cumulative distributions of the gray levels and entropy to adaptively preserve the infrared targets and visible textures. Although these methods produce better subjective effects and higher fusion efficiencies, they use only a single filter to decompose the source image, which results in loss of image features to a certain extent. Yang et al. [15] proposed a multi-scale decomposition method based on a rolling guided filter and a fast bilateral filter to decompose the input images into sublayers. Following this, sparse representations and a detail injection model are used to obtain the fused result with abundant information. Chen et al. [16] proposed a guided filter and multi-directional filter banks, where the filter is used to separate the source image into its base and detail layers while the filter bank is applied to fuse the base layers; this combination was shown to achieve better fusion performance. Luo et al. [17] proposed an IVIF scheme based on visibility enhancement and hybrid multiscale decomposition to obtain the base and detail layers, and the weights of the visual saliency illumination map and a convolutional neural network (CNN) were used to process the corresponding sublayers. Compared to a single filter, the hybrid filter obtains finer details and brighter infrared targets, but the fused image is obtained at the expense of loss of algorithmic efficiency.

The second category includes deep learning (DL)-based methods, which have substantially improved and achieved research results [3,4,9,18–20]. Liu et al. [18] first applied the deep CNN to IVIF to extract features and calculate the activity level to generate feature maps, thereby obtaining fused images. Li et al. [19] constructed an "encoder–fusion-strategy–decoder" framework to achieve high-quality results. Ma et al. [20] employed a generative adversarial network (GAN) for guiding IVIF for the first time, in which the GAN was similar to an adversarial game between an input image and an imaginary image generated from the input image by setting the loss function to continuously adjust the weights to obtain a near-perfect fused image.

To summarize, scale–transform-based methods can obtain sub-layers at different scales with the help of a decomposition algorithm. Thereafter, suitable fusion rules related to image context information are designed to guide the fusion of these sub-layers. However, these methods are inherently flawed as they involve manual comparison, determining the optimal number of decomposition layers, and selecting the fusion rules; the decomposition processing is time-consuming, leading to poor real-time performance. Although DL-based methods have powerful feature extraction functions, they need sufficient raw data and strong computing resources to train models. These methods also lack convincing theoretical knowledge to explain or evaluate the pros and cons of the networks.

Motivated by the abovementioned discussion, we propose an feature-oriented dual-module complementary IVIF. Specifically, we analyzed five classical operators to replace the potential pitfalls of using scale decomposition filters to extract features according to the original image characteristics, and we comprehensively propose two feature extraction modules, namely, the spatial gradient capture module (SGCM) and infrared brightness supplement module (IBSM). As the name suggests, the former is more focused on preserving the spatial gradient information from the original images, which is constructed using principal component analysis (PCA), saliency, and contrast estimation operators. The latter compensates for the feature loss by focusing on improving two exposure metrics that are closely related to image intensity. Extensive experiments were performed on public datasets to prove that the proposed method has better fusion performance in terms of the overall contrast and feature preservation compared to other existing state-of-the-art fusion methods.

The main contributions of this paper can be summarized as follows:

- We propose IVIF via a feature-oriented dual-module complementary. Based on the varying input image characteristics, we analyzed five classical operators to replace the

potential limitations of using scale decomposition filters to extract the features and constructed the two modules, SGCM and IBSM. Owing to the complementarity of these two modules, the fused image shows good performance with adequate contrast and high efficiency.

- We design a contrast estimator to adaptively transfer useful details from the original image, which helps to obtain predicted images with good information saturation. Based on the predicted image, a complementary module is proposed to preserve the color of the visible image while injecting infrared information to generate a realistic fused image.
- We introduce and improve the exposure metrics, namely, the gradient of grayscale (2D gradient) that is responsible for extracting the fine details and well-exposedness for locating the brightness regions. Using these, the infrared information is extracted from the source image and injected into the fused image to highlight the infrared target.

The remainder of this paper is organized as follows. Section 2 briefly introduces the related works. Section 3 describes the proposed IVIF method in detail. Section 4 provides the experimental settings and results analysis, followed by the conclusions in Section 5.

## 2. Related Works

### 2.1. Fast Guided Filter

Assume that $q$ is a linear transform of $I$ in a window $\omega_k$ centered at pixel $k$:

$$q_i = a_k I_i + b_k, \forall i \in \omega_k, \tag{1}$$

where $(a_k, b_k)$ are linear coefficients assumed to be constant in $\omega_k$. This local linear model ensures that $q$ has an edge only if $I$ has an edge because $\nabla q = a \nabla I$.

To determine the linear coefficients, we seek a solution to (1) that minimizes the difference between $q$ and filter input $p$. Specifically, we minimize the following cost function in the window:

$$E(a_k, b_k) = \sum_{i \in \omega_k} \left( (a_k I_i + b_k - p_i)^2 + \epsilon a_k^2 \right), \tag{2}$$

Here, $\epsilon$ is a regularization parameter that prevents $a_k$ from being too large. The solution to (2) is given by linear regression [21] as

$$a_k = \frac{\frac{1}{|\omega|} \sum_{i \in \omega_k} I_i p_i - \mu_k \overline{p}_k}{\sigma_k^2 + \epsilon}, \tag{3}$$

$$b_k = \overline{p}_k - a_k \mu_k, \tag{4}$$

Here, $\mu_k$ and $\sigma_k^2$ are the mean and variance of $I$ in $\omega_k$, respectively, $|\omega|$ is the number of pixels in $\omega_k$, and $\overline{p}_k = \frac{1}{|\omega|} \sum_{i \in \omega_k} p_i$ is the mean of $p$ in $\omega_k$.

After computing $(a_k, b_k)$ for all the patches, known as $\omega_k$, in the image, we compute the filter output as

$$q_i = \frac{1}{|\omega|} \sum_{k:i \in \omega_k} (a_k I_i + b_k) = \overline{a}_i I_i + \overline{b}_i, \tag{5}$$

where $\overline{a}_i = \frac{1}{|\omega|} \sum_{k \in \omega_i} a_k$ and $\overline{b}_i = \frac{1}{|\omega|} \sum_{k \in \omega_i} b_k$.

### 2.2. Well-Exposedness Metric

The well-exposedness feature $(E)$ is initially derived from multi-exposure image fusion (MEIF) proposed by Mertens et al. [22] to preserve well-settled regions in the input

exposed images, i.e., to neglect the under- and over-exposed pixel intensities. Each feature is extracted in the form of a Gaussian curve, whose definition is refined as follows:

$$E_i = exp\left(-\frac{(I_i - 0.5)^2}{2\sigma^2}\right), i = R, G, B, \tag{6}$$

$$E = E_R \cdot E_G \cdot E_B. \tag{7}$$

where "·" represents pixel-wise multiplication, and $\sigma$ is equal to 0.2; $R$, $G$, $B$ are the channels in the exposed image.

## 3. The Proposed Method

This section describes the algorithm framework depicted in Figure 1. First, the input images are fed into the SGCM to obtain abundant spatial feature information. In this module, three operators are proposed for specific roles: PCA is introduced to estimate the overall contour instead of the usual dimensionality reduction, saliency is used to highlight the region of interest, and the contrast estimation operator is proposed by constructing coefficient equations between the source images to adaptively preserve the gradient texture, followed by obtaining the capture maps. Then, a Gaussian–Laplacian pyramid algorithm is used to obtain the predicted image. By comparing the image features between the predicted and raw images, we find that infrared thermal information is lost, and thus propose the IBSM. In the IBSM, two operators applied to the exposure modality are improved by focusing on the image intensity. Then, the supplementary maps are obtained by multiplying the corresponding gradient map and intensity maps estimated via the Sobel gradient operator and Gaussian curve with the given weights. Finally, the fused image is obtained by adding the predicted image to the refined image, calculated by weighting the supplementary maps and source images. The specific steps are as follows.

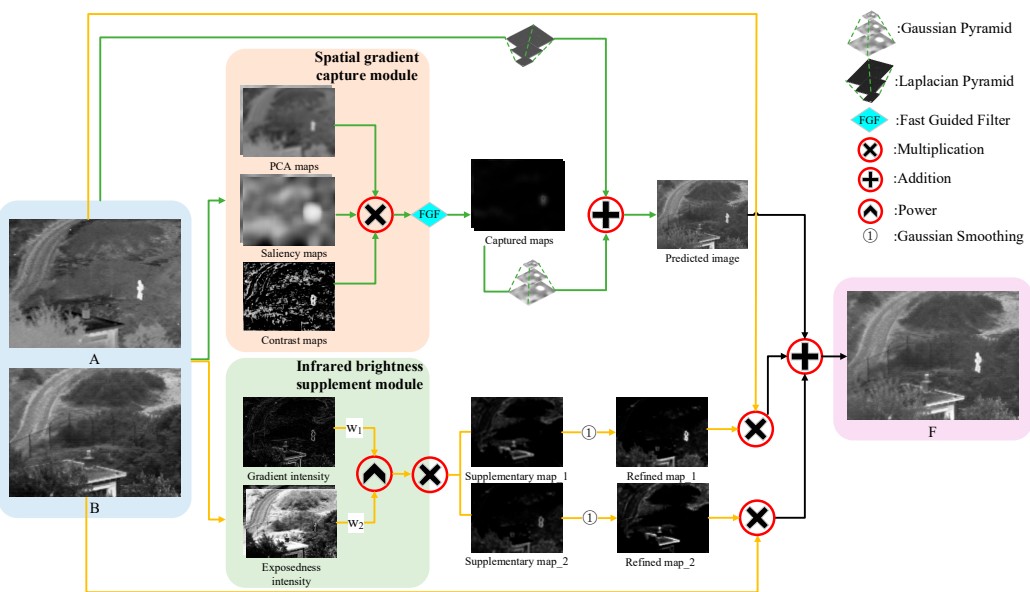

**Figure 1.** Flowchart of the IVIF method via feature-oriented dual-module complementary. Different line formats represent different sections. A, B, and F represent the infrared image, visible image, and fused image, respectively.

### 3.1. Spatial Gradient Capture Module

3.1.1. PCA Operator

Generally, the aim of PCA is to reduce the dimensionality of large datasets by exploiting the underlying correlations between the variables efficiently while preserving most of the information [23]. However, in this paper, PCA is used as a feature extractor to estimate

the weight maps. To the best of our knowledge, PCA has not been utilized for IVIF, but it has already been adopted in MEIF [24].

First, gray-scale images $I_n(n = 1 \ldots N)$ are vectorized into column vectors of size $(\widetilde{r}c \times 1)$, where $\widetilde{r}$ and $c$ denote the numbers of rows and columns of the image, respectively. Then, all these column vectors are combined in a data matrix of size $(\widetilde{r}c \times N)$ consisting of $\widetilde{r}c$ objects having $N$ variables each. After calculating the PCA scores of all objects, each object–variable vector is reshaped to an $(\widetilde{r} \times c)$ image matrix. Next, a Gaussian filter is used to eliminate noise and discontinuities while smoothing the sharp changeovers at the transition regions. Lastly, a sum-to-one normalization is performed at each spatial position $(\widetilde{r}, c)$ over all images, and the final PCA weight map $PCA_n(n = 1 \ldots N)$ is obtained for the fusion operation.

### 3.1.2. Saliency Operator

In image processing, constructing saliency maps is beneficial for observing the human visual system and improving fusion performance. Inspired by Hou et al. [25], an image signature descriptor based on the discrete cosine transform (DCT) is applied to obtain the saliency maps. Given an image $I$, we can approximately isolate the support of the image foreground signal by taking the sign of the DCT of the mixed signal $I$ in the transformed domain and then computing the inverse DCT back into the spatial domain to obtain the reconstructed image $(\bar{I})$. The image signature descriptor (ISD) is defined as

$$ISD(\hat{I}) = sign[DCT(I)], \tag{8}$$

$$\bar{I} = IDCT(ISD(\hat{I})), \tag{9}$$

where $sign(*)$ denotes the entry-wise sign operator.

Subsequently, the saliency maps $(SAL_n)$ are obtained by smoothing the squared reconstructed images, which greatly overlap with the regions of human overt attentional interest and can be measured as the salient points on the input images. Its definition is given as follows:

$$SAL_n = g * (\bar{I} \circ \bar{I}). \tag{10}$$

where $\circ$ and $g$ denote the Hadamard product operator and Gaussian blurring used to suppress the noise introduced by the sign quantization, respectively.

### 3.1.3. Adaptive Contrast Estimation Operator

In general, image contrast reflects the differences in luminance levels between the brightest white and darkest black in the light and dark areas of an image; it is also one of the important elements for measuring the structural details of the image. The magnitude of the image gradient has a low value in the blurred image because the gray-level change at the object edge is not evident. Numerically, the image contrast has a lower value in smooth regions because there are fewer high-frequency components where the grayscale values have large variations. By observing the input images, we found that the visible image often contributes to the contrast distribution of the fused image. However, relying only on the structural details of the visible image and ignoring those in the infrared image inevitably leads to low contrast and poor visual quality of the fused image. To solve this, we propose an adaptive contrast estimation operator to (i) maximize the extraction of the spatial structure details from the infrared image and (ii) preserve the spatial structure of the visible image and the target in the infrared image well, resulting in a contrast map. The specific steps are as follows.

First, calculate the contrast difference map (*CDM*) between the source images as follows:

$$CDM(x,y) = \frac{\max\langle\{AC[IR(x,y)] - AC[VIS(x,y)]\}, 0\rangle}{AC[IR(x,y)]}, \tag{11}$$

where *IR* and *VIS* denote the input images, and $AC[I(x,y)]$ is the contrast of image *I* at the coordinate position $(x,y)$, which is constructed from the original contrast and gradient in the source images [26] as follows:

$$
\begin{aligned}
AC[I(x,y)] = (1-\alpha)(\max I(x',y') - \min I(x',y')) \\
+ \alpha(\max \|\nabla I(x',y')\|),
\end{aligned}
\tag{12}
$$

where $(x',y')$ is the neighborhood pixel of $(x,y)$ within the window size $\mathcal{N}(x,y)$, $\nabla$ denotes the gradient operator, and $\alpha$ is a constant with a value of 0.5.

Thus, by employing *AC* in *CDM* as in Equation (11), the contrast map *CDM* has large values for regions with better spatial details in *IR* compared to *VIS* and low values (or zeros) for the other regions where the spatial details of *VIS* are better. Hence, the adaptive contrast equation inherits the goal that it was built for. Note that the *CDM* comprises simple yet effective calculations for assessing the spatial details of an image and does not require image decomposition via filter banks or frequency decomposition.

After the three different feature maps are extracted from the input images, a fast guided filter (*FGF*) is used to combine them to obtain the captured map (*CM*) as follows:

$$
CM_n = FGF\{(PCA_n. * SAL_n. * CDM), \gamma r, \varepsilon, s\},
\tag{13}
$$

where $r$, $\varepsilon$, and $s$ denote the local window radius, regularization parameter, and subsampling ratio, respectively.

To avoid the appearance of visual artifacts and combine the different scales together, a Gaussian pyramid is constructed for the captured maps as follows:

$$
G_n^i = dg_2\left(CM_n^i\right), \ n = 1, \ldots N; i = 1, 2, \ldots, j.
\tag{14}
$$

where $dg_2(*)$ corresponds to an operator that convolves an image with a Gaussian kernel and then downsamples it to half of its original dimensions; $j$ is the sampling number, and its value is calculated as $j = \text{floor} \{\log [\min(r,c) \ / \ \log(2)]\}$. Then, a set of progressively smaller and smoother weight maps $\left\{G_n^1, G_n^2, \ldots, G_n^j\right\}$ are produced.

Similarly, a Gaussian pyramid is built for each input image $I_n$, and a Laplacian pyramid is constructed for each $I_n$ through the following recursive formula:

$$
L_n^i = I_n^i - ug_2\left(dg_2\left(I_n^i\right)\right),
\tag{15}
$$

where $ug_2(*)$ is an operator used to upsample an image to twice its original size.

Since $L_n^i$ captures the frequency content of the original image at scale $i$, a multi-scale combination of all $I_n^i$ give:

$$
\begin{aligned}
R_n^i &= \left(G_1^1 \cdot L_1^1 + G_N^1 \cdot L_N^1\right) + \left(G_1^2 \cdot L_1^2 + G_N^2 \cdot L_N^2\right) + \ldots + \left(G_1^j \cdot L_1^j + G_N^j \cdot L_N^j\right) \\
&= \sum_{i=1}^{j}\left(\sum_{n=1}^{N} G_n^i \cdot L_n^i\right),
\end{aligned}
\tag{16}
$$

Then, the image $R_n^i$ is reconstructed by upsampling from the high-scale image to the low-scale image to obtain the predicted image $P(x,y)$ as follows:

$$
P(x,y) = \sum_{n=1}^{N}\left(\sum_{i=1}^{j} ug_2\left(R_n^i\right)\right)
\tag{17}
$$

### 3.2. Infrared Brightness Supplement Module

As illustrated in Figure 2, the predicted image has better visual information. Compared with the infrared source image in Figure 2a,b, however, it is observed from Figure 2c,d that the infrared target is dim, which means that the infrared thermal information is not

sufficiently extracted by the previous module. To address this, we propose an IBSM focused on image intensity.

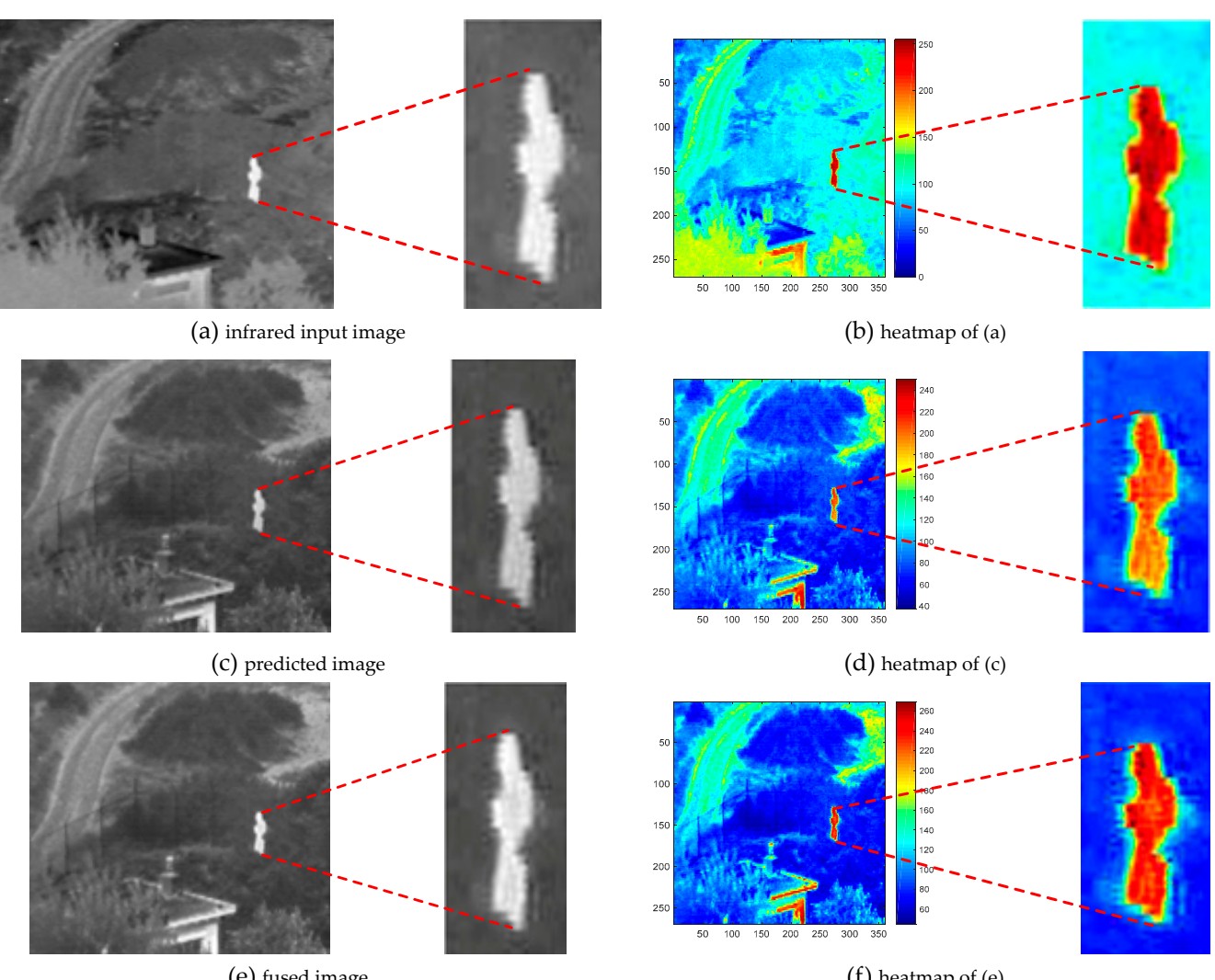

(a) infrared input image

(b) heatmap of (a)

(c) predicted image

(d) heatmap of (c)

(e) fused image

(f) heatmap of (e)

**Figure 2.** Infrared input image, predicted image, fused image, and their corresponding heatmaps.

### 3.2.1. Gradient Intensity Operator

Given a continuous grayscale image $I$, calculate the intensity values in the horizontal and vertical directions. Assume $x$ and $y$ are spatial coordinates such that the intensity components can be denoted by $H(x, y)$ and $V(x, y)$. Then, the grayscale image may be written as $f = [H(x, y), V(x, y)]$. The following notations are adopted: $(x, y) = (x_1, x_2) = x$, $f = (H, V) = (f_1, f_2)$, $y = f(x) = [f_1(x), f_2(x)]$, and $x \in \Re^2$, where $\Re$ is the set of real numbers.

For $i$ and $j = 1, 2$, we assume that the rank of the Jacobian matrix $J = (\partial f_j / \partial x_i)$ is two everywhere in $\Re^2$. Let $f_i(x) = (\partial f_1 / \partial x_i, \partial f_2 / \partial x_i)$. According to this definition, $f_i(x)$ is a two tuple of a real number. Moreover, we postulate that $f_i(x)$ and their first derivatives are continuous. For $i, k = 1, 2$, we set

$$g_{ik}(x) = f_i(x) \cdot f_k(x), \tag{18}$$

where "$\cdot$" is the dot product. According to the above notations, $f_i(x)$ can be written as follows:

$$p = \frac{\partial H}{\partial x} h + \frac{\partial V}{\partial x} v, \tag{19}$$

$$q = \frac{\partial H}{\partial y} h + \frac{\partial V}{\partial y} v, \tag{20}$$

where $h$ and $v$ are unitary vectors associated with $H$ and $V$, respectively. During the calculation of the partial derivatives, two Sobel operators are used as follows:

$$S_x = \begin{bmatrix} 1 & 2 & 1 \\ 0 & 0 & 0 \\ -1 & -2 & -1 \end{bmatrix}, \; S_y = \begin{bmatrix} 1 & 0 & -1 \\ 2 & 0 & -2 \\ 1 & 0 & -1 \end{bmatrix} \tag{21}$$

Similarly, $[g_{ik}(x)]_{i,k=1,2}$ can be represented by

$$g_{xx} = p \cdot p = \left| \frac{\partial H}{\partial x} \right|^2 + \left| \frac{\partial V}{\partial x} \right|^2, \tag{22}$$

$$g_{yy} = q \cdot q = \left| \frac{\partial H}{\partial y} \right|^2 + \left| \frac{\partial V}{\partial y} \right|^2, \tag{23}$$

$$g_{xy} = g_{yx} = p \cdot q = \frac{\partial H}{\partial x} \frac{\partial H}{\partial y} + \frac{\partial V}{\partial x} \frac{\partial V}{\partial y}. \tag{24}$$

In image processing, we are often interested in the following two quantities [27] that are computed locally at each spatial coordinate $(x, y)$: (i) direction through $(x, y)$, along which $f$ has the maximum rate of change; (ii) absolute value of this maximum rate of change. Therefore, we aim to find the maximization of the following form:

$$(df)^2 = g_{xx}dxdx + g_{yy}dydy + g_{xy}dxdy + g_{yx}dydx, \tag{25}$$

under the condition

$$dxdx + dydy = 1.$$

The above problem can also be formulated as finding a $\theta$ value that maximizes the following expression:

$$\underset{\theta}{argmax} g_{xx}cos^2\theta + 2g_{xy}cos\theta sin\theta + g_{yy}sin^2\theta. \tag{26}$$

Let

$$F(\theta) = g_{xx}cos^2\theta + 2g_{xy}cos\theta sin\theta + g_{yy}sin^2\theta, \tag{27}$$

using the common trigonometric function formulas,

$$sin^2\theta = \frac{1}{2}(1 - cos2\theta), \tag{28}$$

$$cos^2\theta = \frac{1}{2}(1 + cos2\theta), \tag{29}$$

$$sin\theta cos\theta = \frac{1}{2}sin2\theta. \tag{30}$$

$F(\theta)$ can be written as

$$\begin{aligned} F(\theta) &= \tfrac{1}{2} \Big[ g_{xx}(1 + cos2\theta) + 2g_{xy}sin2\theta + g_{yy}(1 - cos2\theta) \Big] \\ &= \tfrac{1}{2} \Big[ \big( g_{xx} + g_{yy} \big) + \big( g_{xx} - g_{yy} \big) cos2\theta + 2g_{xy}sin2\theta \Big] \end{aligned} \tag{31}$$

Letting $dF/d\theta = 0$, we obtain

$$\theta(x, y) = \frac{1}{2} arctan \left( \frac{2g_{xy}}{g_{xx} - g_{yy}} \right). \tag{32}$$

Here, $\theta(x,y)$ is the angle that determines the direction through $(x,y)$, along which $f$ has the maximum rate of change. If $\theta_0$ is a solution to this equation, then so is $\theta_0 \pm \pi/2$. As $F(\theta) = F(\theta + \pi)$ on the basis on $tan(\theta) = tan(\theta \pm \pi)$, we may confine the values of $\theta$ to the interval $(0, \pi)$. Thus, Equation (31) provides two values that are $\pi/2$ apart at each $(x,y)$, which means that a pair of orthogonal directions are involved; along one of them, $f$ attains its maximum rate of change, while the minimum is attained along the other. Therefore, the absolute value of this maximum rate of change is given by:

$$G_\theta(x,y) = \left\{ \frac{1}{2} \left[ \left(g_{xx} + g_{yy}\right) + \left(g_{xx} - g_{yy}\right) \cos 2\theta(x,y) + 2g_{xy} sin2\theta(x,y) \right] \right\}^{1/2}, \quad (33)$$

where $G_\theta(x,y)$ denotes the gradient intensity at $(x,y)$.

### 3.2.2. Exposedness Intensity Operator

Exposedness features are often extracted in the MEIF task because it can localize the exposed regions well. Since infrared images are similar to exposure images on the premise of considering the brightness information, this work introduces exposedness to IVIF. However, we also notice that there are two constants (0.5 and 0.2) in Equation (6), which means that the equation does not consider the differences in the source images. This design is actually similar to the commonly used "weighted average" fusion rule, which means that all pixel values are promoted to be equal to 0.5, regardless of the difference in image distribution, while ignoring the regions where the pixel values in the source images are 0 or 1, resulting in loss of structural details and infrared brightness in the fused image. To address this defect, we improved the exposedness intensity ($A_n(x,y)$) by replacing the constants in Equation (6) with the mean and standard deviation to obtain a new equation operator as follows:

$$A_n(x,y) = \exp\left( -\frac{(I_n(x,y) - (1 - \mu_{I_n}))^2}{2\sigma_{I_n}{}^2} \right), \quad (34)$$

where $\mu_{I_n}$ and $\sigma_{I_n}$ are the means and standard deviations of the pixel intensities in $I_n$, respectively. In this case, each exposedness is determined by each input, which can be seen as an adaptive operator.

Next, the supplementary map ($S_n(x,y)$) is obtained through the two feature maps:

$$S_n(x,y) = G_n{}^{\omega_1}(x,y). * A_n{}^{\omega_2}(x,y). \quad (35)$$

where $\omega_1$ and $\omega_2$ are weights that determine the ratio of the two feature maps injected into the fused image.

Then, a Gaussian filter is used to smooth $S_n(x,y)$ to reduce the noise artifacts as follows:

$$R_n(x,y) = \text{Gaussian}(S_n(x,y), \sigma_r), \quad (36)$$

where $\sigma_r$ denotes the standard deviation of the Gaussian kernel.

Finally, the fused image is obtained as

$$F = P(x,y) + \sum_{i=1}^{n} I_i(x,y) \cdot R_i(x,y). \quad (37)$$

Compared to the resulting image in Figure 2c,d, the fused image in Figure 2e achieves better visual results and maintains similar color of the infrared target as that in the infrared input image.

## 4. Experimental Setting and Results Analysis

### 4.1. Experimental Setting

To verify the effectiveness of the proposed method, a large number of experiments are performed on the TNO [28] and RoadScene [29] datasets. Meanwhile, nine state-of-the-art

fusion methods are used to compare the proposed method, including VGG-19 and a multi-layer-fusion-based method (VggML) [30], ResNet and zero-phase component analysis (Resnet50) [31], Bayesian fusion (BayF) [32], algorithm unrolling image fusion (AUIF) [33], classification-saliency-based fusion (CSF) [34], dual-discriminator conditional generative adversarial network (DDcGAN) [35], semantic-aware real-time fusion network (SeAFusion) [36], visibility enhancement and hybrid multiscale decomposition (VEHMD) [17], and Y-shape dynamic transformer (YDTR) [37]. Six indicators are measured for each method as follows: edge-based metric ($Q_{abf}$) [38], structure-based metric (SSIM) [39], multiscale-feature-based metric ($Q_m$) [40], phase-congruency-based metric ($Q_P$) [41], mutual information for the wavelet ($FMI_w$), and discrete cosine ($FMI_{dct}$) features [42]. Higher values of these indicators represent better fusion results.

*4.2. Parameter Discussion*

There are several parameters that need to be discussed to find the optimal values, as shown in Table 1. In this subsection, eight sets of images from the TNO dataset and two groups of images from the RoadScene dataset are averaged for the six objective indicators, and the average value is the largest, and the optimal value is more significant.

**Table 1.** Parameters in the proposed method.

| Parameters | FGF | | | $S_n$ | | $R_n$ |
|---|---|---|---|---|---|---|
| | $r$ | $\varepsilon$ | $s$ | $\omega_1$ | $\omega_2$ | $\sigma_r$ |
| Optimal value | 8 | 0.1 | 2 | 1.0 | 2.8 | 3 |

The role of FGF is to eliminate possible discontinuities and noise in the combined maps. As shown in (13), to determine the optimal parameters, we follow the variable transformation rule, namely, varying one parameter while the others remain fixed. Table 2 shows the average values of the six indicators for different $\varepsilon$, and it is seen that when $\varepsilon$ is 0.1, the averages are the largest. Similarly, we can conclude that the maximum value occurs when $r$ is 8 and $s$ is 2 from the results in Table 3.

**Table 2.** Averages of the six metrics for different $\varepsilon$ on ten pairs of source images from two public datasets. Numbers in bold font represent the best value.

| Metrics | $\varepsilon = 10^{-1}$ | $\varepsilon = 10^{-2}$ | $\varepsilon = 10^{-3}$ | $\varepsilon = 10^{-4}$ |
|---|---|---|---|---|
| $Q_{abf}$ | **0.4894** | 0.4816 | 0.4456 | 0.4091 |
| SSIM | **0.8349** | 0.8270 | 0.7854 | 0.7428 |
| $Q_m$ | **0.7102** | 0.6878 | 0.6246 | 0.5845 |
| $Q_p$ | **0.3974** | 0.3866 | 0.3361 | 0.2891 |
| $FMI_{dct}$ | **0.8924** | 0.8916 | 0.8870 | 0.8821 |
| $FMI_w$ | **0.4146** | 0.4098 | 0.3887 | 0.3713 |

**Table 3.** Averages of the six metrics for different $r$ and $s$ values on ten pairs of source images from two public datasets. Numbers in bold font represent the best value.

| Metrics | $r = 2$ $s = 0.5$ | $r = 2$ $s = 2$ | $r = 4$ $s = 1$ | $r = 4$ $s = 4$ | $r = 8$ $s = 2$ | $r = 8$ $s = 8$ |
|---|---|---|---|---|---|---|
| $Q_{abf}$ | 0.4757 | 0.4710 | 0.4844 | 0.4766 | **0.4920** | 0.4774 |
| SSIM | 0.8270 | 0.8237 | 0.8323 | 0.8269 | **0.8363** | 0.8266 |
| $Q_m$ | 0.6922 | 0.6837 | 0.7049 | 0.6917 | **0.7092** | 0.6858 |
| $Q_p$ | 0.3877 | 0.3851 | 0.3939 | 0.3897 | **0.3994** | 0.3909 |
| $FMI_{dct}$ | 0.8922 | 0.8919 | 0.8924 | 0.8910 | **0.8927** | 0.8909 |
| $FMI_w$ | 0.4096 | 0.4085 | 0.4127 | 0.4115 | **0.4154** | 0.4119 |

As for the parameters $\omega_1$ and $\omega_2$, we know from the above discussion that these determine the importance of the gradient intensity and exposedness strength in the fused image, and the more important features are assigned higher weight ratios. In the IBSM, the focus is more on extracting the infrared heat information missing from the results of the SGCM from the source images and injecting them into the fused images. In other words, we need to ensure that the gradient strength in the fused image remains unchanged when injecting the infrared heat information to realize effective complementation of the two modules. We thus set $\omega_1$ to 1.0, and $\omega_2$ is assigned a higher weight. Through extensive experiments on eight sets of source images and evaluation of the six indicators, the change trend of $\omega_2$ is seen in Table 4 while $\omega_1$ defaults to 1.0. From the table, we find that the average of five of the indicators is at maximum when $\omega_2$ is 2.8, except for $Q_m$. Moreover, the difference between $Q_m$ and the highest value is 0.001, which is a relatively small error. After a final comparison, we assign $\omega_1$ and $\omega_2$ as 1.0 and 2.8, respectively. As observed from Table 5, the average values of the six indicators decrease with continuous increase in $\sigma_r$, which shows that $\sigma_r$ equal to 3 is the most effective.

**Table 4.** Averages of the six metrics for different $\omega_2$ on ten pairs of source images from two public datasets. Numbers in bold font represent the best value.

| Metrics | | $Q_{abf}$ | SSIM | $Q_m$ | $Q_p$ | $FMI_{dct}$ | $FMI_w$ |
|---|---|---|---|---|---|---|---|
| $\omega_1 = 1.0$ | $\omega_2 = 1.0$ | 0.4897 | 0.8336 | 0.7080 | 0.3931 | 0.8922 | 0.4133 |
| $\omega_1 = 1.0$ | $\omega_2 = 1.3$ | 0.4908 | 0.8347 | 0.7090 | 0.3958 | 0.8923 | 0.4140 |
| $\omega_1 = 1.0$ | $\omega_2 = 1.5$ | 0.4912 | 0.8351 | **0.7096** | 0.3969 | 0.8925 | 0.4144 |
| $\omega_1 = 1.0$ | $\omega_2 = 1.8$ | 0.4916 | 0.8357 | 0.7095 | 0.3984 | **0.8926** | 0.4149 |
| $\omega_1 = 1.0$ | $\omega_2 = 2.0$ | 0.4906 | 0.8326 | 0.7059 | 0.3886 | 0.8885 | 0.4134 |
| $\omega_1 = 1.0$ | $\omega_2 = 2.3$ | 0.4919 | 0.8362 | 0.7093 | 0.3994 | **0.8926** | 0.4154 |
| $\omega_1 = 1.0$ | $\omega_2 = 2.5$ | 0.4920 | 0.8363 | 0.7085 | 0.3998 | **0.8926** | 0.4155 |
| $\omega_1 = 1.0$ | $\omega_2 = 2.8$ | **0.4921** | **0.8365** | 0.7086 | **0.4005** | **0.8926** | **0.4158** |
| $\omega_1 = 1.0$ | $\omega_2 = 3.0$ | 0.4920 | **0.8365** | 0.7088 | **0.4005** | **0.8926** | **0.4158** |

**Table 5.** Averages of the six metrics for different $\sigma_r$ on ten pairs of source images from two public datasets. Numbers in bold font represent the best value.

| Metrics | $\sigma_r = 3 * 3$ | $\sigma_r = 5 * 5$ | $\sigma_r = 7 * 7$ | $\sigma_r = 9 * 9$ |
|---|---|---|---|---|
| $Q_{abf}$ | **0.4947** | 0.4879 | 0.4835 | 0.4806 |
| SSIM | **0.8381** | 0.8338 | 0.8310 | 0.8287 |
| $Q_m$ | **0.7106** | 0.7025 | 0.6947 | 0.6900 |
| $Q_p$ | **0.4011** | 0.3987 | 0.3967 | 0.3951 |
| $FMI_{dct}$ | **0.8927** | 0.8921 | 0.8921 | 0.8920 |
| $FMI_w$ | **0.4160** | 0.4152 | 0.4145 | 0.4138 |

### 4.3. Subjective Comparisons

Figure 3 shows the results obtained by ten fusion methods on images from the TNO dataset. In terms of preservation of the infrared brightness information, the six methods based on VggML, Resnet50, BayF, AUIF, CSF, and YDTR yield relatively dim infrared targets, e.g., the persons and wheels in multiple scenes. The fusion results of the DDcGAN, SeAFusion, and VEHMD methods manifest that these can compensate for the abovementioned shortcomings; however, there are some problems that must be noted: the DDcGAN over enhances the contrast of the source images, resulting in the sharpening of both the infrared targets and texture details, and the visible details of SeAFusion are partially lost, e.g., the branches of the trees are intertwined rather than separated, as can be observed from the magnified blue regions. The VEHMD generates undesirable noise and artifacts that alter the image characteristics. In contrast, the proposed method not only maintains the infrared target brightness well but also finely preserves the visible details without noticeable artifacts.

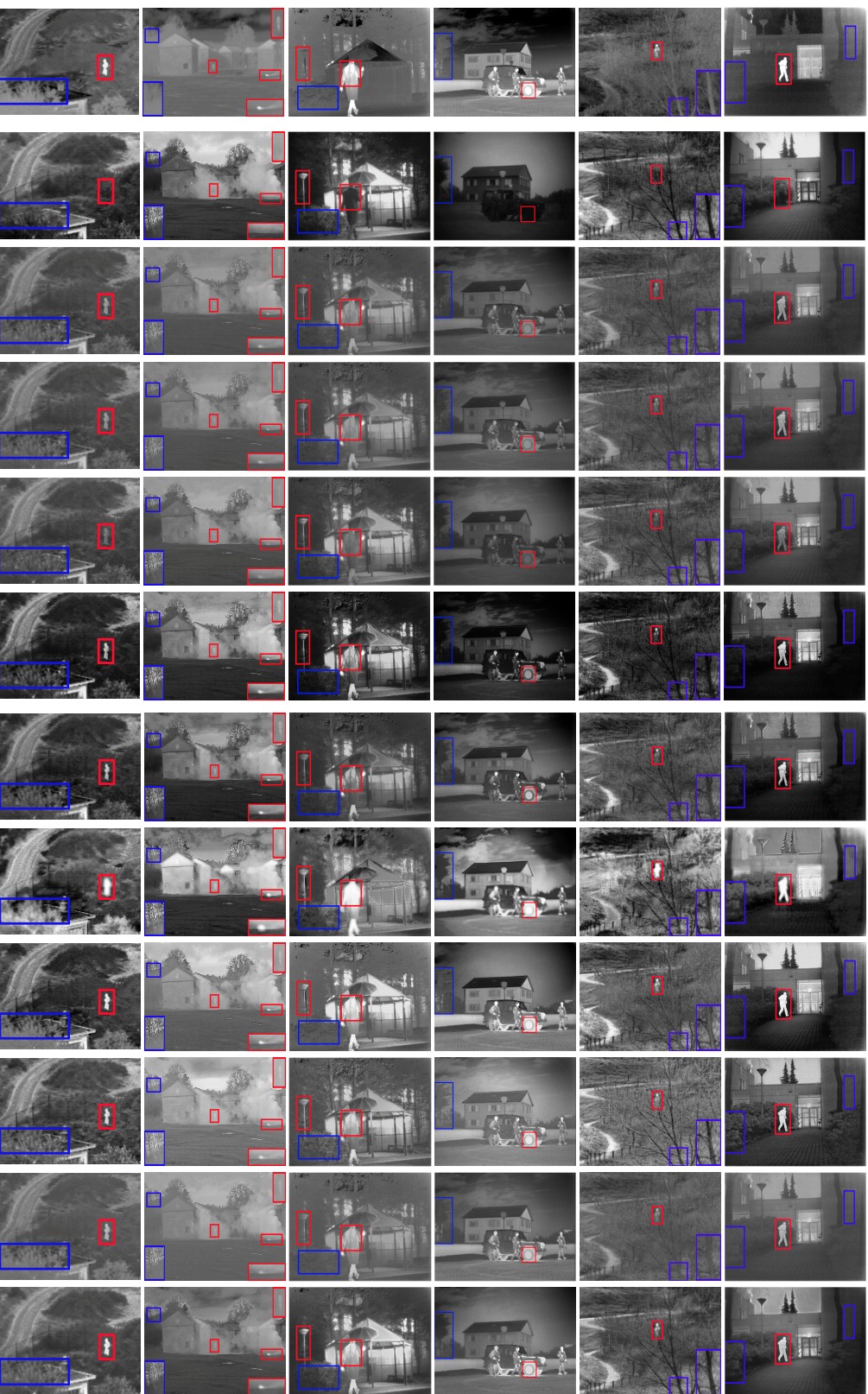

**Figure 3.** Subjective result images of the ten fusion methods. From top to bottom in order: infrared images (IR), visible images (VIS), VggML, Resnet50, BayF, AUIF, CSF, DDcGAN, SeAFusion, VEHMD, YDTR, and our method (Ours). From left to right in order: "Camp", "Octec", "Kaptein 1654", "Jeep", "Sand-path", and "Kaptein 1123". The red and blue boxes show the enlarged local regions.

For further comparison, two groups of image pairs from the RoadScene dataset are examined to obtain the fused results, as shown in Figures 4 and 5. In Figure 4, the overall contrast based on the VggML in Figure 4c, Resnet50 in Figure 4d, BayF in Figure 4e, and VEHMD in Figure 4j are dim, making it difficult to distinguish between infrared brightness information and texture details. The CSF and SeAFusion methods are extreme cases: one is too dim while the other is too bright, and both are fusion results that look unnatural; and the image information of the three lights in the distance is also lost, as shown in the blue enlarged boxes in Figure 4g,i. The DDcGAN in Figure 4h destroys the original texture structure of the source image, resulting in serious artifacts and noise. The AUIF and YDTR methods have improved considerably from the perspective of infrared brightness extraction; the detail information, however, is lost, *e.g.*, the three lights in the blue magnified area in Figure 4f as well as the lightness and texture of the tree trunk in the red magnified area in Figure 4k. By contrast, our method has higher image quality owing to the well-preserved details of the infrared targets and visible features.

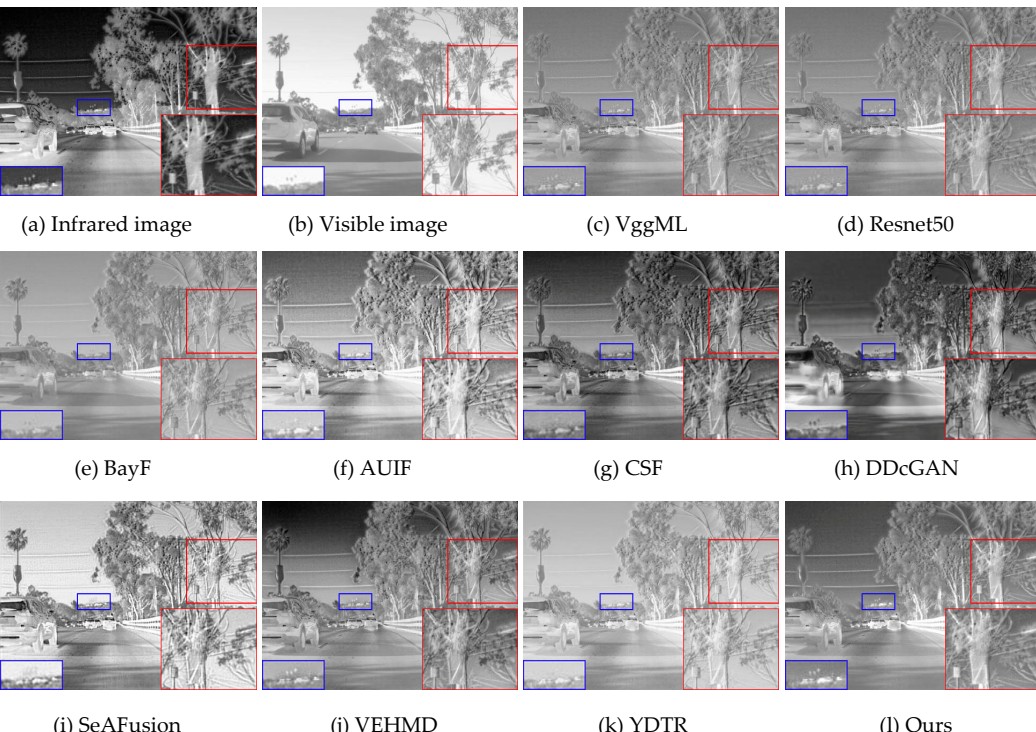

| (a) Infrared image | (b) Visible image | (c) VggML | (d) Resnet50 |
| (e) BayF | (f) AUIF | (g) CSF | (h) DDcGAN |
| (i) SeAFusion | (j) VEHMD | (k) YDTR | (l) Ours |

**Figure 4.** Fused images using ten different methods when the source image is "FLIR_video_00018".

Similar conclusions are observable in Figure 5. The overall contrast of the VggML, Resnet50, and BayF methods are low, resulting in unrecognizable letters on the ground. Although the contrast of AUIF, SeAFusion, and YDTR have improved considerably, the over enhanced contrast makes the fused image look too bright, causing the texture on the wheel to be lost, as can be verified from the blue enlarged areas in Figure 5f,i,k. In addition, the methods based on CSF, DDcGAN, and VEHMD produce artifacts and noise, e.g., the outline of the tree within the blue magnified area. On the contrary, the fused image generated by the proposed algorithm in Figure 5l looks natural and preserves the finer details in the source images.

### 4.4. Objective Evaluation

To evaluate the proposed method more comprehensively, we use the six indicators to test the fusion performances on the TNO and RoadScene datasets containing 30 sets of images, and these results are shown in Figures 6 and 7. From Figure 6, we clearly observe that our method achieves the best average values for all six indicators. Figure 7 demonstrates that our method performs well as it achieves the four highest averages and

two top-three highest averages among the six metrics. Judging from the overall trends of the ten methods on the two datasets, our method has a high probability of exceeding the performances of other state-of-the-art methods.

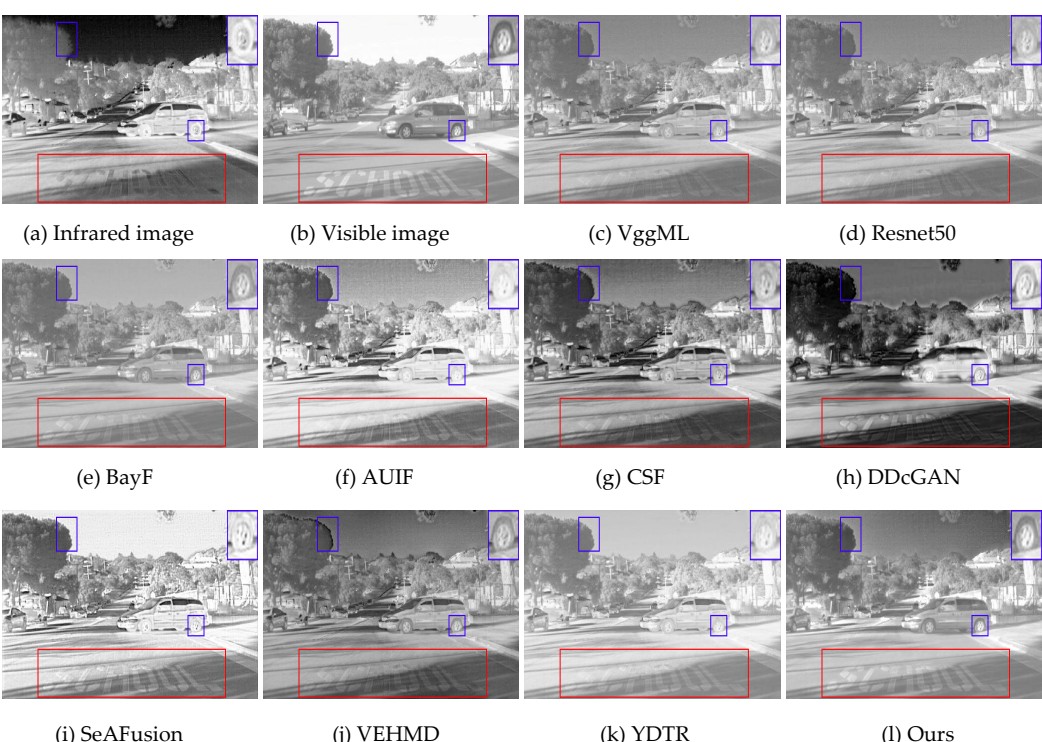

(a) Infrared image      (b) Visible image      (c) VggML      (d) Resnet50

(e) BayF      (f) AUIF      (g) CSF      (h) DDcGAN

(i) SeAFusion      (j) VEHMD      (k) YDTR      (l) Ours

**Figure 5.** Fused images using ten different methods when the source image is "FLIR_video_05245".

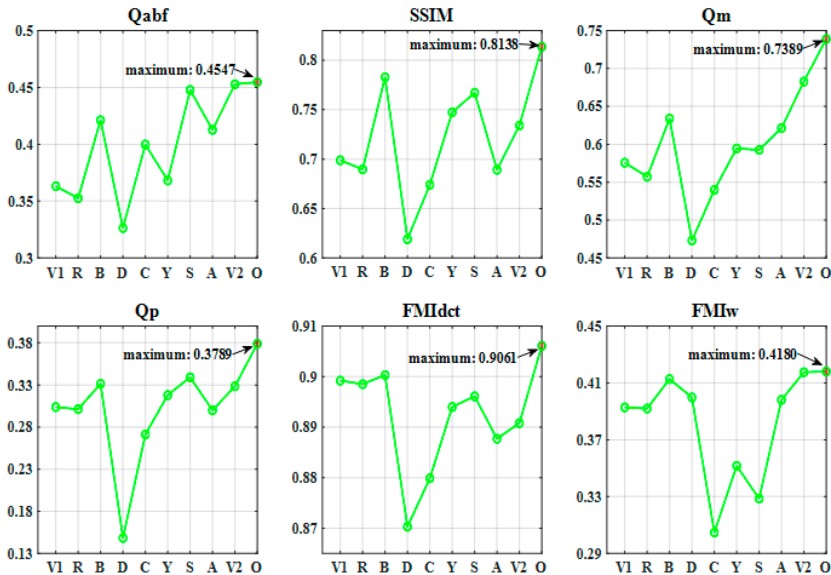

**Figure 6.** Average values of the six objective indicators for 30 sets of images from the TNO dataset. Here, V1, R, B, D, C, Y, S, A, V2, and O denote VggML, Resnet50, BayF, DDcGAN, CSF, YDTR, SeAFusion, AUIF, VEHMD, and Ours, respectively.

### 4.5. Algorithm Effectiveness Analysis

After the dual verification of the subjective effects and objective indicators, we confirm that the proposed method is effective. To reiterate, the proposed method involves building two different yet complementary feature extraction modules based on five typical operators, each of which plays a different role. In this section, therefore, we present a detailed analysis

of the subjective visual maps generated by the mutual promotion of the five operators of the two modules. As shown in Figure 8, we observe that Figure 8(a1,b1) controls the overall contour and Figure 8(c1) reflects that the details are extracted according to scale from small to large and coarse to fine. Their corresponding heatmaps in Figure 8(a2–c2) also represent these feature changes. After the multiplication operation, the captured maps in Figure 8(d1) show integration of each of the feature maps, leading to more uniform image gradient distribution, i.e., the infrared thermal radiation and gradient information is nearly similar to the information distribution of the source images, as verified in Figure 8(d2).

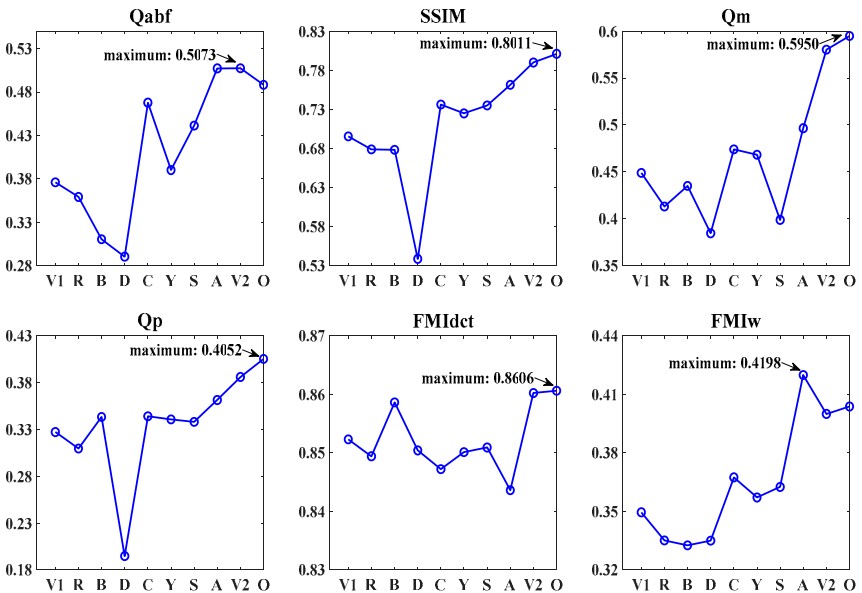

**Figure 7.** Average values of the six objective indicators for 30 sets of images from the RoadScene dataset. Here, V1, R, B, D, C, Y, S, A, V2, and O denote VggML, Resnet50, BayF, DDcGAN, CSF, YDTR, SeAFusion, AUIF, VEHMD, and Ours, respectively.

Similarly, the feature variations of the other module are shown in Figure 9. Unlike the SGCM in Figure 8, which focuses on the spatial gradient features and ignores infrared information, the design of the IBSM in Figure 9 follows two principles: (i) it maintains the overall outline and spatial gradient of the previous module unchanged; and (ii) it extracts infrared information from the source images to the predicted image. Inspired by two metrics related to image exposure, we introduce and improve them in the module to achieve good cross-modal cross-fusion. From Figure 9(a1), the gradient intensity is seen to have clear object outlines and rich infrared brightness, although the visual effect is dim because it is obtained from the horizontal and vertical directions of the source images with the help of the Sobel operator, which has strong edge extraction capability. Subsequently, the exposedness intensity is calculated, and as seen in Figure 9(b1), the brightness features are full. However, maintaining balance between the two intensities when one is too dim and the other too bright is a critical concern. In this case, two weight coefficients $\omega_1$ and $\omega_2$ are used, followed by obtaining supplementary maps. These act similar to a classifier responsible for separating the infrared brightness features from the gradient texture features, as shown in Figure 9(c1,c2). To reduce the noise artifacts, Gaussian smoothing is applied, resulting in refined maps. Finally, the fused images are generated with abundant gradient textures and bright infrared targets.

Through the above analysis, we demonstrate the motivation for each step of the algorithm. It is also seen from the combination of the subjective visual image and objective indicators that the dual-module complementary strategy is successful.

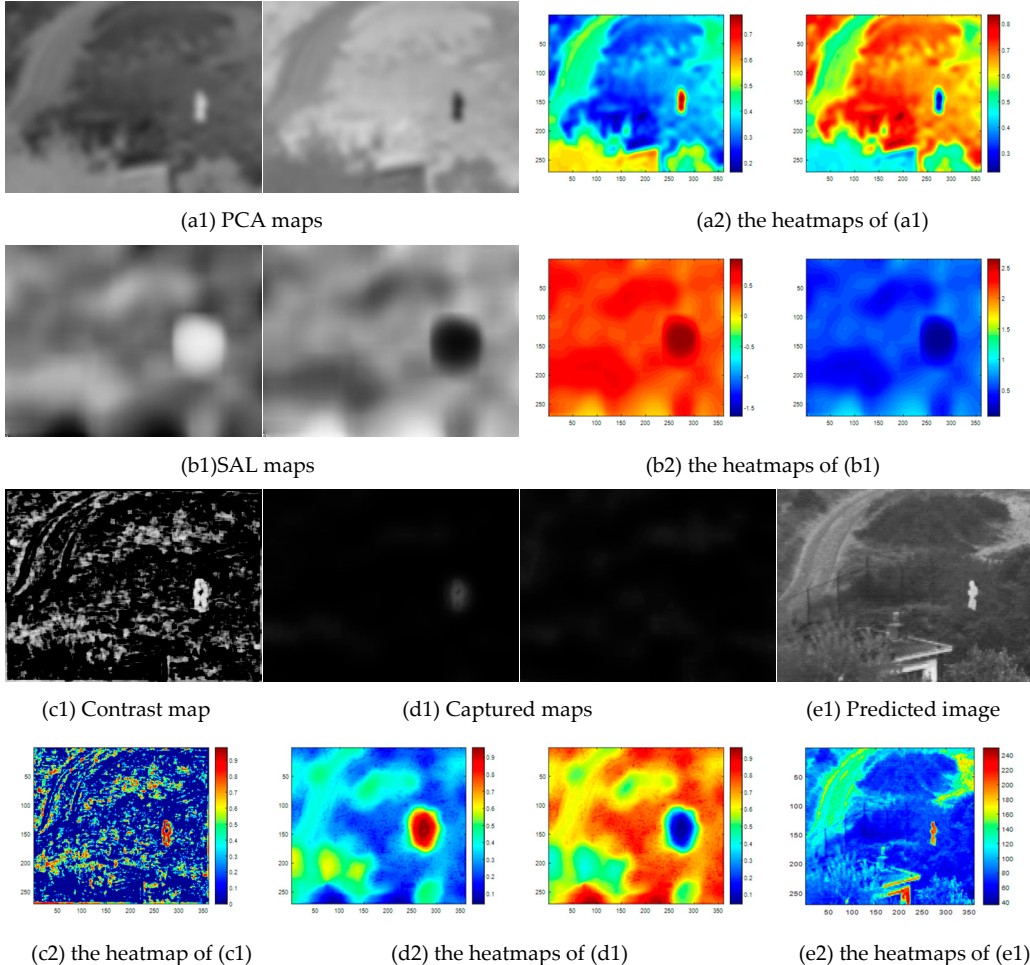

(a1) PCA maps　　　　　　　　　　　(a2) the heatmaps of (a1)

(b1)SAL maps　　　　　　　　　　　(b2) the heatmaps of (b1)

(c1) Contrast map　　　　　(d1) Captured maps　　　　　(e1) Predicted image

(c2) the heatmap of (c1)　　　(d2) the heatmaps of (d1)　　　(e2) the heatmaps of (e1)

**Figure 8.** The visual maps and their corresponding heatmaps of SGCM in the fusion framework (Figure 1).

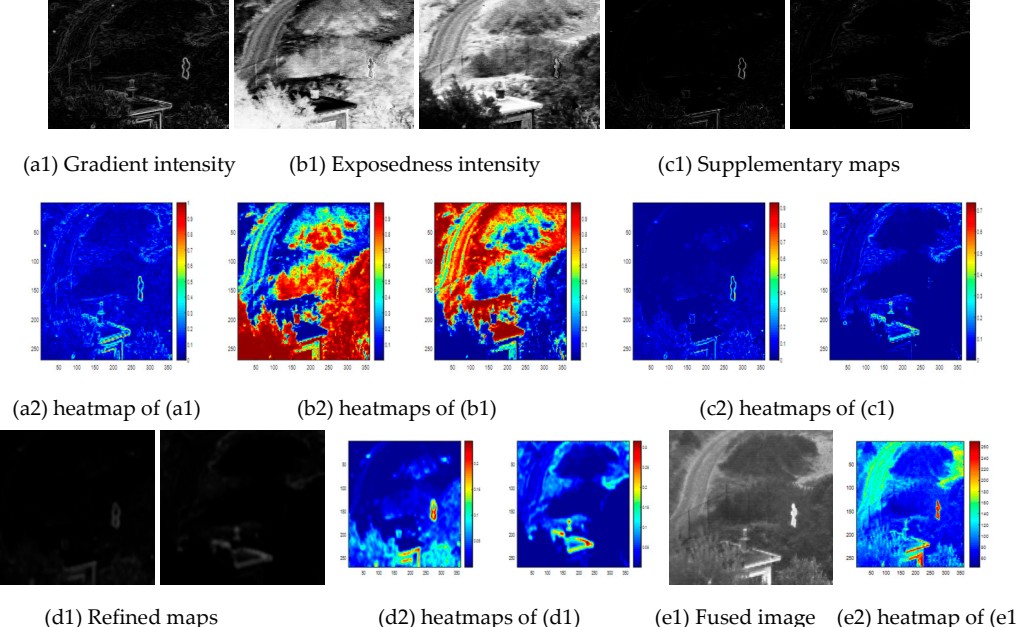

(a1) Gradient intensity　　　(b1) Exposedness intensity　　　(c1) Supplementary maps

(a2) heatmap of (a1)　　　(b2) heatmaps of (b1)　　　(c2) heatmaps of (c1)

(d1) Refined maps　　　(d2) heatmaps of (d1)　　　(e1) Fused image　　　(e2) heatmap of (e1)

**Figure 9.** The visual maps and their corresponding heatmaps of IBSM in the fusion framework (Figure 1).

*4.6. Computational Efficiency*

In order to compare the computational efficiency of the proposed method with other methods, the average running time for a total of 60 images on two image datasets is calculated and presented in Table 6. As shown in Table 6, our method runs faster than CSF, AUIF, and VEHMD. The reason for this is that our method uses five typical operators directly applied to the pixels, while AUIF and VEHMD rely on the time-consuming operation of scale decomposition to obtain various sublayers, resulting in a substantial increase in running time. A few methods require less running time than the proposed method, primarily because the models are pre-trained, including VggML and Resnet50 methods. Despite the effectiveness of these methods, the proposed method yields better results in terms of subjective and objective metrics.

**Table 6.** Average running time on two public datasets.

| Methods | Average Running Time (unit: s) | |
| --- | --- | --- |
| | **TNO Dataset** | **RoadScene Dataset** |
| VggML | 5.9308 | 3.5316 |
| Resnet50 | 3.7758 | 2.7426 |
| BayF | 1.1611 | 0.8939 |
| DDcGAN | 3.2797 | 1.3940 |
| YDTR | 2.1307 | 1.8370 |
| CSF | 14.8467 | 7.7464 |
| VEHMD | 78.2178 | 48.2212 |
| SeAFusion | 0.0033 | 0.0029 |
| AUIF | 12.4612 | 7.1988 |
| Ours | 7.6707 | 7.0148 |

## 5. Conclusions

This paper proposes an effective feature-oriented dual-module complementary IVIF strategy. Unlike the existing multiscale fusion methods with carefully designed decomposition filters to extract features, we focus on cross-modality introduction and improvement of some classic operators to build a fusion framework. First, PCA, saliency, and contrast estimation operators are used to jointly construct a module aimed at obtaining three kinds of feature maps, which are later reconstructed through pyramidal transformation to obtain the predicted image. Then, the IBSM is then proposed to compensate for the missing infrared information in the predicted image by improving the gradient of the grayscale image and well-exposedness, which are measurement operators applied to exposure modalities. The experimental results show that the proposed method has better fusion performance and outperforms other existing mainstream fusion methods. However, the proposed method also has limitations: the reconstruction method uses pyramid transformation, and the number of transformation layers changes adaptively with image resolution, which may increase the running efficiency of the algorithm; these will be improved in a future work.

**Author Contributions:** Y.Z. designed and developed the proposed method, conducted the experiments and wrote the manuscript. H.J.L. designed the new concept, provided the conceptual idea and insightful suggestions to refine it further, and reviewed the manuscript. All authors have read and agreed to the published version of the manuscript.

**Funding:** This research was funded in part by the Basic Science Research Program through the National Research Foundation of Korea (NRF), by the Ministry of Education under Grant 2019R1D1A3A03103736, and in part by the project for Joint Demand Technology R&D of Regional SMEs funded by the Korea Ministry of SMEs and Startups in 2023 (Project No. RS-2023-00207672).

**Informed Consent Statement:** Not applicable.

**Data Availability Statement:** Unavailable due to further research.

**Conflicts of Interest:** The authors declare no conflict of interest.

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
