# Peer review of "Infrared and Visible Image Fusion via Feature-Oriented Dual-Module Complementary"

_applsci, doi:10.3390/app13052907_

Round 1

Reviewer 1 Report

The authors propose an infrared and visible image fusion method in this paper. They construct two modules to highlight the details and incorporate missing infrared thermal information into the predicted image. The paper is well-organized, and the theory is thoroughly demonstrated.

However, I find the figures in the paper unclear. For example,

1)      in Fig. 3, there are many blue and red boxes, but some are hardly visible. My suggestion is that the outlines of the boxes may be thicker, and dash lines would be better.

2)      in Fig. 4, there are magnified views of all boxes indicated in each figure, but in Fig. 5, it seems that only one of those boxes has a magnified view.

3)      even with the magnified views, it is still not very clear to show the enhancement in Fig. 4.

4)      in Fig. 3, although the proposed method adds some missing thermal information, it seems that low-frequency noise is also added. This can be compared among the figures. For example, the results of the proposed method seem much hazier than those of CSF. 

All in all, this paper is interesting, and the proposed method has potential uses.

Author Response

Thank you for your valuable comments. Please see an attached file  for our reply to each comment.

Reviewer 2 Report

Journal : Applied Sciences

Title: Infrared and Visible Image Fusion Via Feature‐Oriented Dual‐Module Complementary

Authors : Yingmei Zhang, Hyo Jong Lee

Review: The manuscript is publishable with minor revisions.

The authors implement a method to quickly and efficiently obtain the fusion between infrared and visible images, combining the results of two complementary modules, one (SGCM) provides better visual information and the other (IBSM) the infrared information that are lost in the SGCM module. The paper is well structured, fluent, understandable and quite reproducible. All the steps of the method have been clearly explained and the results show the efficiency of the tested method, both with a visual approach and with a metric approach.

However, minor revision may be considered:

1)    A PCAnet has been used for IVIF in: Li, S.; Zou, Y.; Wang, G.; Lin, C. Infrared and Visible Image Fusion Method Based on a Principal Component Analysis Network and Image Pyramid. Remote Sens. 2023, 15, 685. https://doi.org/10.3390/rs15030685. The dataset used are the same, what do they think about this paper? Can they compare it to their method?

2)    When the PCA method is introduced they use notation N to indicate the ‘variables’, which variable? Does N correspond to the infrared and visible images (N=2)?

3)    How many PCA components do they use? I suppose the first pricipal component for infrared and visible images.

4)    In Eq. 12, how do they set alpha constant?

5)    In Eq. 13, they wrote FGL, do they mean FGF?

6)    In Eq. 14, does the i index correspond to the gaussian pyarmid layers? Index notation is unclear through the text (confuseable). For example, when they declare j they use r letter I suppose as the row number of the image; it can be confuse with the r local window radius of Eq. 13. 

7)    Why do they use the gaussian pyramid on the raw images and the laplacian one on the captured images? Since they are similar, why this choice? 

8)    Eq. 16 is not well explained and the corresponding part in the flowchart in Fig. 1 is misleading.

When G and L are multiplied, what does the symbol (.) stand for? (Element-wise product ?) Moreover, the subscripts and superscripts in Eq. 16 do not clarify how R is calculated. If it corresponds to the pyramid layer, the images have different sampling, thus how do they sum these?

Eq. 16 is not clearly understandable in the flowchart of Fig. 1, it seems a sum between the gaussian, the captured and the laplacian images.

9)    In Fig. 2. The figures a), b) and c) have the same name. They should change it according to the text.

10) In Eq. 31 when they define F(θ) they miss θ in sin2.

11) Looking at Fig. 8, the captured maps seem to be different from the captured maps in Fig.1 flowchart, where they remind the saliency maps.

12) When they discuss the choice of the parameters in Section 4.2, they use eight sets of images just from the TNO dataset. If they use the images of the RoadScene dataset can the parameters change? And why did they choose those ranges for the parameters and those combination between r and s? (Table 3)

13) Did they estimate the time computational efforts?

Author Response

(The authors gave the same response as above.)
